# Poor Diet, Long Sleep, and Lack of Physical Activity Are Associated with Inflammation among Non-Demented Community-Dwelling Elderly

**DOI:** 10.3390/healthcare10010143

**Published:** 2022-01-12

**Authors:** Maria Basta, Christina Belogianni, Mary Yannakoulia, Ioannis Zaganas, Symeon Panagiotakis, Panagiotis Simos, Alexandros N. Vgontzas

**Affiliations:** 1Department of Psychiatry, University Hospital of Heraklion, 71500 Heraklion, Greece; belogianni.christina@gmail.com (C.B.); akis.simos@gmail.com (P.S.); avgontzas@psu.edu (A.N.V.); 2Sleep Research and Treatment Center, Department of Psychiatry, Penn State University, State College, PA 16802, USA; 3Department of Nutrition and Dietetics, School of Health Science and Education, Harokopio University of Athens, 17671 Athens, Greece; mary.yannakoulia@gmail.com; 4Department of Neurology, University Hospital of Heraklion, 71500 Heraklion, Greece; johnzag@yahoo.com; 5Department of Internal Medicine, University Hospital of Heraklion, 71500 Heraklion, Greece; simeongpan@hotmail.com

**Keywords:** inflammatory markers, elderly, objective sleep, diet, physical activity

## Abstract

Inflammation in elderly is associated with physical and cognitive morbidity and mortality. We aimed to explore the association of modifiable lifestyle parameters with inflammation among non-demented, community-dwelling elderly. A sub-sample of 117 patients with mild cognitive impairment (MCI, *n* = 63) and cognitively non-impaired controls (CNI, *n* = 54) were recruited from a large, population-based cohort in Crete, Greece, of 3140 elders (>60 years old). All participants underwent assessment of medical history/physical examination, extensive neuropsychiatric/neuropsychological evaluation, diet, three-day 24-h actigraphy, subjective sleep, physical activity, and measurement of IL-6 and TNFα plasma levels. Associations between inflammatory markers and diet, objective sleep duration, subjective sleep quality, and lack of physical activity were assessed using multivariate models. Regression analyses in the total group revealed significant associations between TNF-α and low vegetable consumption (*p* = 0.003), and marginally with objective long nighttime sleep duration (*p* = 0.04). In addition, IL-6 was associated with low vegetable consumption (*p* = 0.001) and lack of physical activity (*p* = 0.001). Poor diet and lack of physical activity appear to be modifiable risk factors of inflammation, whereas long sleep appears to be a marker of increased inflammatory response in elderly. Our findings may have clinical implications given the association of inflammatory response with morbidity, including cognitive decline, and mortality in elderly.

## 1. Introduction

In the elderly, several molecular and cellular changes of the innate and acquired immunity have been described as potential contributors to biological aging [1]. Among others, dysregulation of the immune system, known as inflammaging, is characterized by low-grade but constant elevations of pro-inflammatory markers, such as interleukin (IL)-1, IL-6, IL-8, IL-13, C-reactive protein (CRP), interferon a (IFNa) and interferon b (IFNb), and tumor necrosis factor-alpha (TNFα). High levels of pro-inflammatory markers are evident in the majority of elderly, even among those without risk factors or clinically active diseases [2]. Pro-inflammatory responses, cellular senescence and immune senescence [3,4] are important components in the disorders of aging, including cardiovascular disease, cancer, chronic kidney disease, dementia, and depression, and other age-related conditions, such as functional decline, sarcopenia, frailty and mortality [2]. In light of these observations, there is a growing trend to approach overall mortality and morbidity, through the study of how various modifiable factors, such as diet, sleep and physical exercise, can affect the inflammatory load and consequently impact overall quality of life [5].

In older populations, research examining the relation of dietary parameters and inflammatory markers is rather scarce and has not produced consistent results so far. Observational and intervention studies, performed in either community-dwelling or hospitalized older individuals, concluded that sufficient evidence exists only regarding the protective role of n-3 polyunsaturated fatty acid intake [6,7]. In relation to diet overall, two studies in older adults from Italy and Scotland found that closer adherence to a Mediterranean diet, a plant-based dietary pattern, is related to decreases in inflammation-related markers over time (three or six years, respectively) [8,9]. However, little is known about the associations between inflammatory markers and specific food groups (i.e., vegetables, red meat, dairy, legumes, etc.) in older adults, including elderly with mild cognitive impairment (MCI), information that may be easily addressed in daily clinical practice.

Other studies have focused on possible links between sleep and inflammation in older adults. Sleep architecture changes significantly as we age [10,11], and health decline contributes to increasing sleep problems among the elderly, such as insomnia, sleep apnea, circadian rhythm sleep–wake disorders, and poor sleep quality and quantity [10,12]. Research so far indicates a U-shape association between objective sleep measurements and increased inflammatory markers among cognitively non-impaired elderly (CNI) [13,14,15,16], while others found no significant association between them [17,18].

Finally, in relation to physical activity, there is solid evidence reporting that physical activity based on amount and intensity of leisure time activities are related to lower concentrations of inflammatory markers in blood in older population groups [19,20,21,22,23,24,25,26]. Notably, this association has not been systematically studied using simple questions such as frequency of daily continuous brisk walking, reflecting a common and accessible for elderly form of physical activity with evidence for health benefits based on the UK Chief Medical Officer’s guidelines [27].

In terms of exercise, higher levels based on rather strenuous exercise programs have a beneficial effect on inflammation levels in healthy elderly [22]. Longitudinal studies investigating associations between inflammatory markers and exercise in patients with MCI are very few. A recent pilot study demonstrated that 12 weeks of aerobic training in 30 CNI and 30 patients with MCI significantly decreased inflammatory markers in both groups [28]. Moreover, a recent meta-analysis has linked lifestyle interventions, including exercise, with significant reductions in IL-6 and TNFα levels in patients with MCI and dementia [29].

MCI is an early stage of impairment in memory and/or other cognitive functions (such as language, executive, or visuoconstruction capacity) in individuals who maintain the ability to independently perform most activities of daily living and sustain a normal level of function [30,31,32]. On the other hand, the prevalence of MCI is high (about 15–20% in elderly above 60 years), with a significant annual rate in which MCI progresses to dementia varying between 8 and 15% per year [31]. Based on that, MCI is an important condition to identify and treat. Therefore, identifying possible modifiable factors that may delay or reverse the progression of MCI to dementia is very important.

To our knowledge, there is no study evaluating the joint contribution of all three parameters of lifestyle, i.e., diet, sleep, and physical activity, as described above in relation to inflammation markers in older non-demented adults, including patients with MCI. To fill this gap in the literature, the aim of the present study was to examine in a comprehensive way the association between inflammation and diet based on food groups, subjective and objective sleep and mild physical activity based on daily continuous, brisk walking, in a sample of community-dwelling older adults. A secondary objective was to explore if the purported role of each of the three lifestyle parameters is modified by the presence of age-related cognitive impairment as indicated by diagnosis of MCI. Based on our previous work and existing literature, we hypothesized that objective sleep, diet, and physical activity will independently relate to inflammation levels, both in cognitively non-impaired elderly, as well as among persons with MCI.

## 2. Materials and Methods

### 2.1. Study Design

The present sample consisted of participants of the Cretan Aging Cohort, a cross-sectional study of community-dwelling elders, recruited from the rural and urban areas in the district of Heraklion, Crete, Greece, between March 2013 and June 2015. The primary aim of this study was to investigate the prevalence of and risk factors associated with cognitive decline [33]. The Cretan Aging Cohort study was conducted in two phases. The study was conducted according to the guidelines laid down in the Declaration of Helsinki and all procedures involving human subjects/patients were approved by the Bioethics Committee of the University Hospital of Heraklion, Crete (Protocol Number: 13541, 20 November 2010). Written informed consent was obtained from all subjects.

Phase I: Eligible participants were those aged > 60 years who visited selected primary health care facilities in areas of the Heraklion district for any reason. Consenting individuals (*n* = 3200) completed an interview with a specially trained nurse, who used a structured questionnaire to document sociodemographic information, anthropometric measurements, physical and mental health issues, and medication use. Cognitive function was evaluated using the Greek version of the Mini Mental State Examination (MMSE) test [34], applying a universal cut-off of 23/24 points (because the majority of participants had ≤6 years of formal education) for referral of patients for further evaluation. Based on this cut-off, participants were divided into two groups: those with MMSE <24, considered to be at risk for cognitive impairment, and the not-at-risk group with MMSE ≥24 [33]. After excluding participants with crucial missing data (MMSE score, age), the final study sample consisted of 3140 people (57.0% women) aged 73.7 ± 7.8 (60–100) years, who had completed an average of 5.8 ± 3.3 (0–18) years of formal education and lived mostly in rural areas.

Phase II: Participants who scored < 24 points on the MMSE (*n* = 636) were referred for an extensive neuropsychological and neuropsychiatric evaluation in phase 2 of the study. The 344 consenting subjects were similar with the 636 originally refereed subjects in terms of age, gender, and body mass index (BMI). Certified neurologists, psychiatrists and internists completed an extensive questionnaire based on the one used in the Hellenic Longitudinal Investigation of Aging and Dietstudy [35]. Trained neuropsychologists performed a test battery evaluating a variety of cognitive domains. Apart from medical and family history, daily activities, dietary patterns and sleep characteristics were assessed as well (for a detailed description of all tests and scales used, see [33]). Diagnosis of any type of MCI was based on modified Petersen criteria (IWG-1) [36] and on a consensus decision between two or more clinicians who took into account results from the comprehensive neuropsychiatric and neuropsychological evaluation. Diagnosis of MCI further required that cognitive deficits could not be accounted for by clinically significant mood or anxiety disorder.

To be included in the MCI group participants had to have age- and education-adjusted z scores <−1.5 on indices derived from at least two tests within a given cognitive domain (episodic memory, language, attention/executive) and demonstrate intact levels of every-day functionality according to a comprehensive, informant scale of instrumental activities of daily living [37] adapted for the Greek population from Lawton and Brody (1969). Using the non-cognitively impaired pool of subjects who scored >24 on the MMSE (*n* = 2504), a control group of 181 participants was created after stratifying for residence, gender, and age. Of those, 161 agreed to participate in phase 2 of the study (see Figure 1). In phase II, among 505 participants examined, 231 were diagnosed as MCI, 128 with dementia, and 146 were CNI [33].

### 2.2. Participants

The current analyses were performed on a sub-sample of participants from Phase II in whom we obtained complete data of objective (actigraphy) and subjective sleep, measurements of inflammatory markers (TNF-α and IL-6 plasma level), as well as dietary and physical activity habits based on valid questionnaires. The final sample included non-cognitively impaired participants (CNI) (*n* = 54), and participants with a diagnosis of MCI (*n* = 63) (Figure 1). Given that all participants in this cohort were community-dwelling members of primary care health units, elderly people with severe terminal medical illnesses or severe movement impairment were not included in this cohort.

### 2.3. Measurements

#### 2.3.1. Inflammatory Markers

Blood samples were collected from each participant between 10:00 am and 12:00 pm, transferred to EDTA-containing tubes (three per participant) and refrigerated until centrifugation (within 3 h) for plasma isolation. Afterwards, the plasma samples were kept in deep freeze (−80 °C). Plasma TFN-α and IL-6 were measured by ELISA technique (Human TNF-alpha Quantikine HS ELISA and Human IL-6 Quantikine HS ELISA kits respectively, R&D Systems Europe, Abington, UK). For the TNF-α measurement, the inter-assay coefficient of variation was 12.74%, the intra-assay coefficient of variation was 19.04, and the lower detection limit was 0.209 pg/mL. For the IL-6 determination, the inter-assay coefficients of variation were 13.09%, the intra-assay coefficients of variation were 11.04, and the lower detection limit was 0.133 pg/mL.

#### 2.3.2. Diet

Habitual diet was assessed using a validated semi-quantitative food frequency questionnaire [38]. It comprises 69 questions on the consumption of foods or combination of foods, including dairy products, cereals, fruits, vegetables, meat, fish, legumes, added fats, alcoholic beverages, stimulants and sweets. Using a 6-point scale (“never/rarely”, “1–3 times/month”, “1–2 times/week”, “3–6 times/week”, “1 time/day”, “≥2 times/day”), participants were asked to indicate the absolute frequency of consuming a certain amount of food, expressed in g, millilitres, or in other common measures, such as slice, tablespoon or cup, depending on the food. Responses to the food frequency questionnaire were converted to daily intakes of specific food items and were extrapolated into macronutrient intakes. Energy intake was calculated by summing energy intake from macronutrients and alcohol, assuming 4 kcal/g for carbohydrates and proteins, 9 kcal/g for lipids, and 7 kcal/g for alcohol. Responses were also grouped into major groups (expressed as servings/day). Adherence to the Mediterranean dietary pattern was evaluated using the MedDietScore, an 11-item composite score calculated for each participant from the FFQ-based food consumption [39]. The score is based on the weekly consumption of 11 food groups (non-refined cereals, fruits, vegetables, legumes, potatoes, fish, meat and meat products, poultry, full fat dairy, olive oil use and alcohol). A score 0–5 is given for each food group. The potential range of MedDietScore is between 0 and 55, with higher values indicating greater adherence to the Mediterranean diet.

#### 2.3.3. Sleep Measurements

Subjective Sleep: Subjective sleep and presence of any sleep disorder were evaluated using 20 sleep-related questions from a standardized questionnaire, used in the Penn State Cohort and described in detail elsewhere [40]. In the current analysis, subjective sleep variables examined were non-refreshing sleep, excessive daytime sleepiness (EDS), sleep apnea symptoms and insomnia symptoms qualified in terms of severity on a scale of 0–4 (0 = none, 1 = mild, 2 = moderate, 3 = severe). Non-refreshing sleep was evaluated by a positive response (“often” or “always”) to the following question “Do you feel groggy and un-refreshed after you wake up in the morning”. Excessive Daytime Sleepiness (EDS) was evaluated by a positive response (“often” or “always”) to one or both of the questions: “Do you feel groggy or sleepy most of the day but manage to stay awake”, and/or “Do you have irresistible sleep attacks”. The presence of symptoms consistent with sleep apnea was indicated by a positive response on one or both of the following questions: “Do you know/Have you been told that you stop breathing or breath irregularly during sleep, occasionally, often or always?” and “Do you know/Have you been told that you snore during sleep to a moderate/severe degree?”. Finally, the presence of “insomnia symptoms” was established on a “yes” answer to the questions: “Do you have difficulty falling asleep”, “Do you have difficulty staying asleep” or “Do you wake up in the morning earlier than desired” often or always.

Objective Sleep: Objective measurement of sleep in a free-living environment was documented for three consecutive 24-h periods during weekdays using an actigraph (Actigraph, GT3XP model, Pensacola, FL, USA) placed on the non-dominant arm of the participant. The actigraphic data were examined together with sleep diaries in which the actigraph was removed and “bedtime”/“out of bedtime” times were reported daily. Night was defined based on the questions “What time did you go to bed” and “What time did you get out of bed” answered by the participant in the sleep diary. Naps, defined as daytime sleep for more than 20 min, were based on the actigraphy and the sleep diary provided by the participant [41,42]. Actigraphy data including among other night sleep efficiency (SE), night sleep onset latency (SOL), night total sleep time (TST), night time in bed (TMB), night wake time after sleep onset (WASO), number of awakenings during the night, night average duration of awakenings, 24-h (nighttime and daytime) sleep time (24-hTST), and 24-h (nighttime and daytime) time in bed (24-h TMB) were analyzed using the ActLife 6 software (Actlife v6.9.5 LLC, Pensacola, FL, USA). Periods without actigraphic activity which, based on the sleep diaries, were not identified as sleep were considered as artifacts and were removed from the analysis. Sleep parameters were averaged over three consecutive 24-h. Participants with actigraphy recorded over fewer than three days or average night TST ≤ 3 h were excluded from the analysis. The start time of the 24-h period was set at 11:00 am on the day that the actigraph was applied on the participant’s hand by our staff. Objective sleep parameters were analyzed as continuous variables in all group comparisons. From the objectively recorded sleep time data, we regrouped the entire study sample into two ordinal groups. Initially, the total group was divided in quartiles. Our previous findings based on the same cohort have shown that long but not short sleep duration is associated with cognitive impairment both among CNI and MCI participants. Based on that, we divided the sample to the top 25% of persons above the median percent sleep time (long sleep duration group), and the 75% of persons in the bottom half (normal sleep duration group). We then rounded the cut-off points to meaningful numbers and thus created the following two sleep duration groups: the long sleep duration group consisted of those who slept ≥7.5 h (coded as 1), and the normal sleep duration group of those who slept <7.5 h (coded as 0) [43].

Finally, short sleep duration was defined as total night <360 min, corresponding to the lower quartile of the distribution of night total sleep time (TST) in the total sample of the present study.

#### 2.3.4. Physical Activity

Regular physical activity was based on participant responses to a single question “How many days did you walk for more than 10 min in a row in a brisk manner during the last week”. Since we aimed to evaluate the role of mild physical activity, lack of physical activity was defined if participants had less than three days of 10 min of brisk walking in a row during the last week, given that adults engaging in less than 30 min of activity per week are considered as significantly inactive [44]. Lack of activity was coded with a value of 1 and presence of activity with 0.

#### 2.3.5. Other Variables

Demographic variables such as gender, and age, and body mass index (BMI) were also assessed. Additionally, prescription of any type of psychotropic medication (including benzodiazepines) was recorded. Finally, a diagnosis of late-life depression was based on the Phase II clinical interview by a team of a certified psychiatrist, a neurologist and a geriatrician, score on self-report psychiatric symptom scales, and the thorough neuropsychological evaluation.

### 2.4. Statistical Analysis

In preliminary univariate analyses we assessed diagnostic group differences on demographic variables, sleep, and lifestyle habits and proinflammatory cytokines. Group differences on continuous variables (total sleep time, sleep efficiency, sleep latency, wake time after sleep onset, cytokine levels, dietary intake) were assessed through ANOVAs and on categorical variables (subjective sleep indices, physical activity) using chi-square tests. In these analyses age, gender, BMI, use of psychotropic medications, sleep apnea symptoms, and diagnosis of depression served as covariates.

The main objective of the study involved assessing the joint contribution of demographic, sleep (objective and subjective), and lifestyle variables on IL-6 or TNF-a levels in a sample of elders varying considerably in cognitive status. Variable selection for multivariate models was based on partial correlations, computed in the total sample, between each independent variable and IL-6 or TNF-a levels, controlling for MCI diagnosis (coded as 0 for the NCI group and 1 for the MCI group). Variables of interest were entered in the multivariate models if their corresponding partial correlation with IL-6 or TNF-a had a *p*-value < 0.1. These multiple linear regression models included IL-6 or TNF-a as the dependent variable and age, gender, ΒΜΙ, and MCI diagnosis as confounders. Both models (the first with IL-6 and the second with TNF-a as the dependent variable) were recomputed, stratified by clinical group (with age, gender, and BMI as confounders). The effects of interest in these analyses were (i) R^2^ change (evaluated at study-wise *p* < 0.05/6 = 0.008), and (ii) the regression coefficients for each independent variable entered in the second step of each model (following all covariates), which were evaluated at family-wise *p* < 0.05/3 = 0.017 for the model predicting TNF-a and at *p* < 0.05/2 = 0.025. A priori power estimation (using G*Power 3.1 [45]) indicated that the sample size was adequate to detect a minimum change in R^2^ = 0.095 (corresponding to a small effect size) to ensure power of 80% at *p* = 0.05.

## 3. Results

### 3.1. Sample Demographics, Clinical and Preclinical Variables

The final sample included 117 participants (63.7% women; Table 1) aged 74.5 years (SD = 7.4), who had attained an average of 5.1 (SD = 3.1) years of formal education. The sample average MMSE score was 24.9 points (SD = 3.6) and comprised 54 cognitively non-impaired (CNI) and 63 persons diagnosed with MCI (Figure 1). Table 2 shows inflammatory markers, diet, subjective and objective sleep variables, and physical activity in the total sample (corresponding values by clinical group are presented in the Appendix A). In univariate analyses, the two clinical groups were comparable on levels of IL-6/TNF-a, sleep parameters, frequency of mild activity (*p* > 0.1) and the majority of dietary variables with some exceptions (descriptives by clinical group and corresponding effect sizes are presented in Table 3). Specifically, CNI participants reported greater adherence to the Mediterranean diet, greater daily energy intake, and greater consumption of red meat, non-refined and refined cereals, and sweets, compared to MCI participants.

### 3.2. Associations of Inflammatory Markers with Sleep, Diet and Physical Activity

Univariate analyses: As shown in Table 4, in the entire sample, higher TNF-α levels were significantly associated with age (r = 0.230, *p* = 0.017), nighttime TST over 450 min (r = 0.202, *p* = 0.03), less frequent consumption of vegetables (r = −0.407, *p* < 0.001) and red meat (r = −0.190, *p* = 0.03). These correlations were adjusted for MCI diagnosis, which was weakly associated with inflammatory markers (Spearman r < 0.13). Moreover, higher levels of IL-6 were significantly associated with lack of mild physical activity (r = −0.303, *p* = 0.002) and less frequent consumption of vegetables (r = −0.393, *p* < 0.001). Correlations of inflammatory markers with all other variables (including depression diagnosis) were also very small (Spearman r < 0.15).

A similar pattern of associations between pro-inflammatory markers and food groups/physical activity was observed within each group, although the association between TNF-a and red meat consumption failed to reach significance, probably due to reduced statistical power (*p* > 0.4).

Multivariate analyses: According to the results of the univariate analyses, the following variables of interest were included in the model accounting for TNF-a levels: long nighttime TST, consumption of vegetables, and consumption of red meat. Entered together in the second step of the model (following age, gender, BMI, and MCI diagnosis which were entered as confounders in the first step), these variables resulted in significant model improvement (change in R^2^ = 0.168, *p* < 0.001). As shown in Table 5, high TNF-α levels were significantly associated with lower vegetable consumption (*p* = 0.001) and marginally with night TST of more than 450 min (*p* = 0.04). In the analyses stratified by MCI diagnosis, consumption of vegetables remained significant in both groups (*p* < 0.016), although long nighttime TST failed to reach significance (*p* > 0.2).

The following variables of interest were entered in the model accounting for IL-6 levels in the total sample: physical activity and consumption of vegetables jointly accounted for 24.2% of variance beyond that explained by step 1 confounders (age, gender, ΒΜΙ, MCI diagnosis; change in R^2^ = 0.246, *p* < 0.001). As shown in Table 6, higher levels of IL-6 were significantly associated with lack of physical activity (*p* = 0.001) and less frequent consumption of vegetables (*p* = 0.001). In the analyses stratified by MCI diagnosis, consumption of vegetables remained significant in both groups (*p* < 0.009), while lack of physical activity failed to reach significance (*p* = 0.06). Post-hoc power estimation indicated that the present sample size ensured adequate power (80%) to reveal a statistically significant independent contribution of physical activity to IL-6 levels (at *p* < 0.05), modest power (56–63%) to reveal statistically significant contribution of vegetable and meat consumption to TNF-a levels, and much lower power (26–36%) regarding the contribution of long nighttime TST and vegetable consumption to TNF-a and IL-6 levels, respectively.

## 4. Discussion

The main findings of this study are that the plasma levels of pro-inflammatory cytokines, i.e., TNF-α and IL-6, among community-dwelling elderly with and without mild cognitive impairment are independently and significantly associated with decreased consumption of vegetables, objective long sleep duration and self-reported lack of physical activity. It appears that poor diet and lack of physical activity may be predisposing factors related with inflammation in elderly, whereas long sleep duration seems to be a marker of increased inflammation in this population.

The first key finding of this study is that a diet low in vegetables correlates with high levels of both TNF-α and IL-6. This correlation was significant in the entire elderly group, as well as both in the cognitively non-impaired and the MCI patients’ subgroups, separately. Previous literature has examined associations between Mediterranean diet and inflammation in various age groups, however few studies have reported associations between inflammatory markers and specific food groups in elderly populations. Specifically, some studies in young-middle-aged or wide age range cohorts reported associations between inflammatory markers such as CRP, IL-6, IL-1β and TNFα, TNFαR2 and vegetables /or vegetables and fruit combined [48,49,50,51,52]. However, none of these studies were focused on the elderly. One previous study in healthy, elderly men failed to find associations between vegetables consumption and CRP [20]. This study showed that in non-demented elderly, IL-6 and TNFα were significantly associated with low consumption of vegetables. Our finding may be explained by the special characteristics and environment of our sample, i.e., rural population, residing in a region in the Mediterranean Sea with mild sunny climate, and local produce of a large variety of vegetables and fruit, which were easily accessible, cheap and part of daily diet consumed in rather high amounts. It is known that the Cretan cuisine makes wide use of a large variety of greens and vegetables; based on previous research, Cretans consume up to three times more vegetables compared to other Europeans [53,54]. Moreover, based on previous literature, vegetables intake in Crete is high compared to other cohorts of older adults in western countries [55,56,57], but also in Greece [58].

Furthermore, to our knowledge, this is the first study to report an association between pro-inflammatory cytokines, IL-6 and TNFα and low vegetable consumption in elderly with mild cognitive impairment. Previous literature reports elevation of pro-inflammatory cytokines, IL-6 and TNFα in patients with degenerative dementias [59,60,61], while other studies have shown that adherence to Mediterranean diet, rich in specific food groups including vegetables, may be beneficial to cognitive function [62,63]. Future interventional studies in larger samples, focusing on the elderly with or without MCI, are needed to validate and expand our findings.

The second significant finding is that long objective total sleep time (TST) > 450 min is associated with higher levels of TNF-α in non-demented community-dwelling elderly. This effect reached significance after correcting for multiple comparisons in the subgroup of persons with MCI and approached significance in the analyses on the total sample. Although a recent meta-analysis including both objective and subjective sleep measurements in a wide age-range of adults [64] also reported an association between long sleep and inflammation, to our knowledge, only one previous original study reported this association in elderly men without cognitive impairment [65]. Our study confirms and expands previous findings on the association between long-sleep and higher inflammatory levels in elderly populations.

A study based on objective actigraphic sleep found a U-shape association between sleep duration and all-cause mortality among elderly [66]. Several studies have reported associations between long sleep duration and cardiovascular disease, coronary heart disease, cognitive decline, and all-cause mortality in this age group [43,67,68,69,70]. It appears that objective long sleep is a result/marker of increased inflammation in elderly subjects [65]. In contrast, and at the other end of the spectrum, short sleep duration, either habitual or as a result of experimental sleep deprivation, appears to induce higher levels of inflammatory markers both in elderly [14] and younger populations [71,72,73,74,75]; a condition that is reversed by lengthening sleep duration [76,77]. The underlying mechanisms relating long sleep and increased inflammation involve the pro-inflammatory cytokines IL-1β and TNFα, which have a significant role in the regulation of sleep both in animals and humans [78,79]. In addition, elevated cytokine levels are associated with excessive sleep and daytime sleepiness. Previous literature has demonstrated that exogenous administration of IL-1β and IL-6 to patients can induce somnolence and/or increased sleep [79]. Furthermore, in patients with excessive daytime sleepiness disorders, such as sleep apnea and narcolepsy, peripheral levels of IL-6 and TNFα are elevated and appear to mediate sleepiness and fatigue [80]; indeed, in a pilot study, Etanercept, a TFNα antagonist, significantly and markedly reduced daytime sleepiness in patients with sleep apnea [81].

Finally, we found that lack of physical activity is correlated with increased IL-6 levels. Many studies in the past have shown the inverse association of physical activity/exercise and inflammation in elderly populations [20,21,22,23,24,26,82,83,84]. In some of these studies physical activity based on self-reported amount and intensity of leisure activities or exercise programs appeared to be strenuous [20,21,22,23,24,84], whereas others used objective measures of physical activity such as step count, hand grip and accelerometry, and the chair stand test among others [24,26,82,83]. Furthermore, this study including MCI patients further confirms and expands earlier findings involving determinants of proinflammatory cytokines among these patients, i.e., associations between exercise and IL-6 and TNFα among patients with MCI with co-morbid insulin resistance, and significant decrease of inflammatory markers, such as IL-6, TNFα and CRP, in MCI patients after a 12-week exercise program [28]. This study showed that even mild self-reported activity assessed with a single question easily administered in a primary care setting, i.e., walking for a minimum of 10 min in a row in a brisk manner on three or more days per week, may have a significant impact on systemic inflammation in both cognitively intact and MCI elderly.

### 4.1. Strengths and Limitations

The main strength of the current study is the inclusion of multiple, modifiable lifestyle factors such as diet, sleep, and physical activity as potential correlates with underlying inflammation in elderly. Furthermore, our study has included a fairly large sample living in an area with unique dietetic habits such as large consumption of fresh vegetables. In addition, our outcome variables are easily measurable and applicable in a primary care setting such as single questions or actigraphy. Finally, the associations between diet, physical activity and inflammation were significant in the entire group and remained significant or borderline significant in both subgroups, i.e., cognitively intact or patients with MCI.

However, some limitations should also be acknowledged. In our study, actigraphy was performed for three consecutive 24-h periods, compared to 1–2 weeks of actigraphy recordings which is the recommended duration. Additionally, we used only self-reported symptoms of apnea to screen for obstructive sleep apnea (OSA), a condition highly prevalent among the elderly [85]. Similarly, EDS and subjective sleep duration were based on participants’ self-reports. Additionally, in our model we did not include other factors potentially associated with inflammation, such as medication use and medical co-morbidities. Moreover, physical activity was based on self-report. However, the fact that vegetables are significantly related with both inflammatory markers examined in the regression models supports their independent role in inflammation in our sample. Finally, because of the cross-sectional nature of this study, causality between inflammation and lifestyle variables such as diet, sleep and physical activity cannot be examined.

### 4.2. Implications

Inflammatory response is associated with morbidity, including cognitive decline and mortality in elderly. Our study extends the existing literature by linking in a comprehensive way poor vegetable consumption, long sleep duration, and lack of physical activity with higher levels of inflammation among non-demented community-dwelling elderly. We propose that adopting a diet rich in vegetables and a physically active lifestyle, by lowering inflammation, could help reduce mortality and morbidity, including cognitive decline among elderly. Furthermore, given that long sleep seems to relate with inflammation levels, as well as cognition and disease severity among elders with cognitive decline [42,43,86], we and others have previously reported that treatment with sedative/sleep-prolonging psychotropic medication in this population should be given with caution [42,43,86,87]. These simple practical recommendations could be easily applied in a primary care setting.

## Figures and Tables

**Figure 1 healthcare-10-00143-f001:**
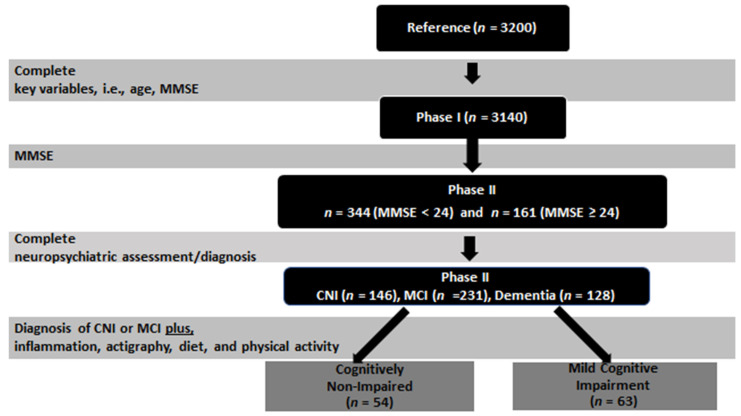
Study flowchart. Abbreviations; MMSE: Mini Mental State Examination, CNI: cognitively non-impaired, MCI: mild cognitive impairment.

**Table 1 healthcare-10-00143-t001:** Demographic and clinical parameters of the total sample and by diagnostic group.

	Total Group|(*n* = 117)		CNI (*n* = 54)		MCI (*n* = 63)			
	Mean	SD	Mean	SD	Mean	SD	*p*	Cohen’s d
Age (years)	74.0	7.4	72.3	7.6	75.6	7.0	0.005	0.45
Education (years)	5.1	3.1	5.4	2.7	4.8	3.5	0.3	0.19
Gender (%)							0.3	0.18
Men	36.3	-	40.7		28.6			
Women	63.7	-	59.3		71.4			
MMSE	24.9	3.6	27.0	2.8	22.9	2.8	<0.001	1.19
GDS	3.5	3.4	2.7	3.4	4.5	3.3	0.008	0.50
HADS-Anxiety	2.8	3.3	2.6	3.4	3.4	3.3	0.3	0.25
Depression (%)	27.4	-	11.1	-	39.7	-	0.001	0.65
Psychotropic medication (%)	31.9	-	24.2	-	44.2	-	0.011	0.53
Benzodiazepine use (%)	8.8	-	9.7	-	8.2	-	0.9	0.11
BMI	30.0	4.8	30.5	4.9	29.3	4.5	0.06	0.26

MMSE: Mini Mental State Examination; GDS: Geriatric Depression Scale; HADS: Hospital Anxiety and Depression Scale. ΒΜΙ: body mass index, CNI: cognitively non-impaired, MCI: mild cognitive impairment. Note: Corresponding values for the combined, total sample of CNI and MCI participants in the Cretan Aging cohort (*n* = 377, Mean age = 75.4, SD = 7.3) are as follows. MMSE: Mean = 24.5, SD = 3.5; GDS: Mean = 3.7, SD = 3.5; HADS-Anxiety: Mean = 3.0, SD = 3.4; BMI: Mean = 30.1, SD = 4.7. Total CNI group; MMSE: Mean = 26.4 (SD = 3.1); GDS: Mean = 3.1 (SD = 3.2); HADS-Anxiety: Mean = 2.9 (SD = 3.6); BMI: Mean = 30.5 (4.7). Total MCI group; MMSE: Mean = 22.3 (SD = 2.6); GDS: Mean = 4.3 (SD = 3.5); HADS-Anxiety: Mean = 3.3 (3.2); BMI: Mean = 29.9 (SD = 4.6).

**Table 2 healthcare-10-00143-t002:** Inflammatory markers, sleep, diet, and physical activity levels of the entire sample (*n* = 117).

	Mean	SD		Mean	SD
IL-6 (pg/mL)	1.28	0.9	Energy	2228.1	536.20
TNF-α (pg/mL)	1.12	0.6	MDS	34.7	4.6
Night TST (min)	411.2	70.9	Servings per day of:		
24-h TST (min)	448.7	84.2	Vegetables	2.54	1.28
Night TST > 450 min (%)	21.0	-	Red meat	1.27	0.84
Night TST < 360 min (%)	26.1	-	Dairy	1.06	0.80
Night TMB (min)	503.2	78.8	Legumes	0.65	0.45
24-h TMB (min)	560.4	107.5	Non-refined cereal	0.91	0.94
Night Sleep Efficiency	83.0	8.8	Refined cereal	2.56	1.20
Night WASO	77.0	42.3	Potatoes	0.38	0.24
Night Sleep latency	13.0	12.0	Fruit	2.83	1.95
Number of Awakenings	15.9	6.1	Fish	0.83	0.66
Sleep duration (min)	394.0	115.9	Poultry	0.52	0.37
Non-refreshing sleep (%)	12.9	-	Eggs	0.19	0.19
Leg movement (%)	1.7	-	Sweets	0.44	0.45
EDS (%)	2.6	-	Alcoholic beverages	0.54	0.97
Sleep Apnea symptoms (%)	17.2	-			
Insomnia-type symptoms (%)	32.5	-			
Physical activity (%)	64.0	-			

Note: Consumption of major food categories is in servings/day. TST: total sleep time, TMB: time in bed, WASO: wake time after sleep onset, EDS: excessive daytime sleepiness, Εnergy: total energy intake (kcal/day), MDS: Mediterranean diet score. IL-6 normal levels < 2 pg/mL [46], TNFα normal levels < 3.1 pg/mL [47].

**Table 3 healthcare-10-00143-t003:** Inflammatory Markers, diet, sleep, and physical activity by diagnostic group.

	CNI (*n* = 54)	MCI (*n* = 63)	*p*	Cohen’s d		CNI (*n* = 54)	MCI (*n* = 63)	*p*	Cohen’s d
IL-6 (pg/mL)	1.36 (1.0)	1.21 (0.7)	0.7	0.17	Energy Intake	2455.8 (472.2)	2017.9 (508.7)	0.2	0.89
TNF-α (pg/mL)	1.07 (0.6)	1.16 (0.6)	0.1	0.16	MDS	35.9 (4.3)	33.6 (4.6)	0.016	0.49
Night TST (min)	408.9 (77.8)	414.5 (64.6)	0.4	0.09					
Night TST > 450 min (%)	16.1	25.0	0.2	0.24	Servings per day of:				
Night TST < 360 min (%)	25.8	27.3	0.9	0.02	Vegetables	2.72 (1.30)	2.41 (1.22)	0.7	0.24
24-h TST (min)	438.0 (92.9)	457.8 (75.3)	0.2	0.24	Red meat	1.47 (0.95)	1.12 (0.68)	0.002	0.55
Night TMB (min)	493.4 (88.1)	511.6 (68.3)	0.14	0.25	Dairy products	1.08 (0.80)	1.05 (0.82)	0.5	0.04
24-h TMB (min)	539.8 (117.9)	578.0 (95.9)	0.1	0.36	Legumes	0.64 (0.40)	0.66 (0.49)	0.8	0.05
Night Sleep Efficiency	82.9 (8.0)	81.2 (9.3)	0.3	0.22	Non-refined cereals	1.14 (1.08)	0.73 (0.78)	0.3	0.44
Night WASO	71.1 (41.3)	82.0 (43.2)	0.3	0.25	Refined cereals	2.65 (1.16)	2.48 (1.24)	0.2	0.13
Night Sleep latency	11.2 (7.0)	14.5 (14.7)	0.1	0.28	Potatoes	0.38 (0.23)	0.40 (0.25)	0.3	0.08
Number of Awakenings	15.4 (6.5)	16.3 (5.8)	0.9	0.15	Fruit	3.14 (1.77)	2.57 (2.07)	0.2	0.34
Sleep duration (min)	412.3 (123.1)	378.0 (106.6)	0.5	0.29	Fish	0.89 (0.74)	0.78 (0.58)	0.5	0.17
Non-refreshing sleep (%)	11.1	14.5	0.5	0.16	Poultry	0.51 (0.36)	0.53 (0.39)	0.7	0.06
Leg movement (%)	1.9	1.6	0.9	0.02	Eggs	0.24 (0.22)	0.15 (0.14)	0.017	0.47
EDS (%)	1.6	3.7	0.6	0.16	Sweets	0.54 (0.48)	0.35 (0.41)	0.006	0.43
Sleep Apnea symptoms (%)	16.7	17.7	0.9	0.04	Alcoholic beverages	0.74 (1.21)	0.38 (0.69)	0.1	0.39
Insomnia-type symptoms (%)	27.8	36.5	0.3	0.23					
Physical activity (%)	62.0	65.6	0.9	0.03					

Notes: Consumption of major food categories is in servings/day. Unless otherwise indicated, values are means (SD), CNI: cognitively non-impaired, MCI: mild cognitive impairment, TST: total sleep time, TMB: time in bed, WASO: wake time after sleep onset, EDS: excessive daytime sleepiness, Εnergy: total energy intake (kcal/day), MDS: Mediterranean diet score.

**Table 4 healthcare-10-00143-t004:** Partial correlations of demographic, clinical, sleep, and lifestyle parameters with proinflammatory cytokine levels in the entire sample.

	TNF-α	IL-6
Age	0.230 †^1^	0.157
Gender (male)	0.157	0.147
Depression Diagnosis	−0.164	−0.081
Physical activity	−0.070	−0.303 †^5^
Night TST > 450 min	0.202 *^2^	0.079
Insomnia Symptoms	−0.131	−0.043
Consumption of vegetables	−0.407 †^3^	−0.393 †^6^
Consumption of red meat	−0.190 *^4^	−0.140
Consumption of dairy	0.142	0.059
Consumption of legumes	−0.049	−0.134

Note: Pearson or Spearman r values controlling for Mild Cognitive Impairment diagnosis. Variables are listed if the zero-order correlations with either IL-6 or TNF-a were associated with *p* < 0.1. TST: total sleep time. * *p* < 0.05, † *p* < 0.01, ^1^ CNI: r = 0.192, *p* = 0.2; MCI: r = 0.245, *p* = 0.06, ^2^ CNI: r = 0.288, *p* = 0.04; MCI: r = 0.069, *p* = 0.5, ^3^ CNI: r = −0.418, *p* < 0.001; MCI: r = −0.328, *p* = 0.01, ^4^ CNI: r = −0.144, *p* = 0.3; MCI: r = −0.100, *p* = 0.4, ^5^ CNI: r = −0.318, *p* = 0.02; MCI: r = −0.298, *p* = 0.02, ^6^ CNI: r = −0.402, *p* = 0.002; MCI: r = −0.369, *p* = 0.003.

**Table 5 healthcare-10-00143-t005:** Associations of TNF-α with modifiable lifestyle habits using Multiple Linear Regression analysis in the entire sample.

	B	95% CI	*p* Value
Night TST > 450 min	0.230	(0.010 to 0.449)	0.04 ^1^
Consumption of vegetables	−0.203	(−0.289 to −0.117)	0.001 ^2^
Consumption of red meat	−0.102	(−0.077 to 0.282)	0.3
Constant	0.595	(−0.741 to 1.931)	0.4

TST: total sleep time; variables were included in the final models if they correlated with TNF-a at *p* < 0.1. Confounders: age, gender, ΒΜΙ, Mild Cognitive Impairment diagnosis. Change in R^2^ = 0.174, *p* < 0.001. Unstandardized regression coefficients are reported throughout, in bold if significant at Bonferroni-adjusted *p* < 0.017. ^1^ CNI: B = 0.287, *p* = 0.2; MCI: B = 0.130, *p* = 0.2, ^2^ CNI: B = −0.240, *p* = 0.002; MCI: B = −0.152, *p* = 0.016.

**Table 6 healthcare-10-00143-t006:** Associations of IL-6 with modifiable lifestyle habits using Multiple Linear Regression analysis in the entire sample.

	B	95% CI	*p* Value
Physical activity	−0.538	(−0.847 to −0.230)	0.001 ^1^
Consumption of vegetables	−0.300	(−0.428 to −0.172)	0.001 ^2^
Constant	1.887	(−1.169 to 3.943)	0.07

Variables were included in the final models if they correlated with IL-6 at *p* < 0.1. Confounders: age, gender, ΒΜΙ, Mild Cognitive Impairment diagnosis. Change in R^2^ = 0.242, *p* < 0.001. Unstandardized regression coefficients are reported throughout, in bold if significant at Bonferroni-adjusted *p* < 0.025. ^1^ CNI: B = −0.739, *p* = 0.06; MCI: B = −0.315, *p* = 0.09, ^2^ CNI: B = −0.303, *p* = 0.008; MCI: B = −0.222, *p* = 0.009.

## Data Availability

The data presented in this study are available on request from the corresponding author.

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
