# Peer review of "Poor Diet, Long Sleep, and Lack of Physical Activity Are Associated with Inflammation among Non-Demented Community-Dwelling Elderly"

_healthcare, 2022, doi:10.3390/healthcare10010143_

Round 1

Reviewer 1 Report

Thank you for the opportunity to read this interesting manuscript. The current investigation is relevant since it adds to the literature about the importance of modifiable risk factors for dementia and poor aging – a fact that we all are experiencing. The risk factors investigated were diet, physical activity, and sleep. There is an imperative need in the scientific literature of investigations on the theme in diverse populations.

Some minor and major concerns are presented:

Abstract

Reflects the content of the manuscript.

Introduction

Minor corrections:

Page 1

Line 33 – [ ] is empty, reference is missing. Please, correct.

Line 36 – IFNa and IFNb must be extensively written, as you did for the others before abbreviation: interleukin (IL)…

Line 38 – [ ] is empty, reference is missing. Please, correct.

Line 40 – no need for an abbreviation of CVD as you do not use it again in the manuscript. Please exclude.

There is a confusion of the terms “physical activity”, “physical exercise” and “exercise” throughout the manuscript. Please, define the Introduction or Methods section and be consistent.

In the abstract, you write physical activity, at Line 44 Page 2 you write physical exercise, and at Lie 374 Page 10 you write exercise.

They are not the same thing; they are quite different concepts.

“Physical activity is defined as any bodily movement produced by skeletal muscles that result in energy expenditure. The energy expenditure can be measured in kilocalories. Physical activity in daily life can be categorized into occupational, sports, conditioning, household, or other activities. Exercise is a subset of physical activity that is planned, structured, and repetitive and has as a final or an intermediate objective the improvement or maintenance of physical fitness. Physical fitness is a set of attributes that are either health- or skill-related. The degree to which people have these attributes can be measured with specific tests”. From: PMID: 3920711

Page 2

Line 44 - At the end of paragraph one you state that “[…] modifiable factors, such as diet, physical exercise and sleep […]”

The second paragraph  (Line 46) is about diet, the third (Line 58) should be on physical activity, and the fourth on sleep (Line 65). Please change the order (of the paragraphs or the sentence).

Line 50 – What is n-3 PUFA intake? n-3 polyunsaturated fatty acids and whey proteins? Please write extensively.

Line 55 - Please delete the sentence: “Furthermore, to our knowledge … MCI”, as at the end of the Introduction you already wrapped it up in a good manner. This is repetitive.

Line 59 – please delete the word “which”: “sleep architecture changes sig…”

Line 60 – I suggest adding: “[…] problems among elderly, such as insomnia sleep apnea, circadian rhythm sleep-wake disorders, and poor sleep quality and quantity.” Exclude the word “symptoms”, leaving only insomnia, as it could mean symptoms and disorder.

Line 62 – the first time that cognitively non-impaired elderly appears, please abbreviate CNI.

I think it's easier for the reader instead of CNI written through the manuscript, just write controls.

Lines 58-64 – please add a recent reference (2020 or 2021 from meta-analysis or systematic reviews)

Line 69 – what does “mild physical activity” means? (you mention it later again on line 80)

Line 72 – please use the abbreviation CNI or controls, as you already have written before

Again in this paragraph (Lines 65 to 75), you misunderstood the terms exercise and physical activity. You can cite exercise, but you must explain that your work is with physical activity and state the literature on the theme or the lack of it.

Methods

Page 4

Line 135 – you mention “valid questionnaires”. But later you describe simple questions asked to your participants, and not valid questionnaires in all measures. Please be consistent.

Page 5

Line 174 – I could not find the detailed description of the sleep questionnaires as you indicated in reference # Bixler EO, Vgontzas AN, Lin HM, Vela-Bueno A, Kales A. Insomnia in central Pennsylvania. J Psychosom Res. 2002 Jul;53(1):589-92. doi: 10.1016/s0022-3999(02)00450-6. PMID: 12127176.

Line 176 – You cannot determine the presence of PLM by a question. That is not correct. You can, instead, investigate the self-perception of PLM, but I don't think the participants would know the disease. So, If I move excessively during sleep I might have PLM? This could lead to bias.

Line 194 – Please add a reference to the sentence: “Naps, defined as....”, as I have never seen a consensus definition of a nap. So if the participant slept for 15 min this was not considered a nap, based on what?

Page 6

Line 203 – sleep parameters were based on 24h for 3 days. Do you have objective data on the physical activity then? Why did you not use it?

Line 213 – Physical activity was defined by walking 10 mins non-stop for 7 days? Did I understand correctly? Where is the reference?

I think at this point you have a serious flaw.

Here is also the place that you need to explain “mild physical activity” mentioned anteriorly.

Results

Page 7

Line 252 -Figure, with F

Line 256 – here you mentioned “frequency of mild exercise” – you did not mention that you assessed exercise of your sample, and specified the type, frequency, and intensity of exercise. I think you mentioned physical activity.

Line 261 - Table 1 and supplementary tables: I strongly suggest having the scores for MMSE, GDS, HADS, and BMI for this age range of your sample.

Line 264 - Table 2: I think it would be interesting to have “normal data” of the actigraphy and IL-6 and TNF data at the bottom of the legend, as the reader doesn’t have a parameter to compare the numbers provided at the table.

How did you measure “non-refreshing sleep”?

What does the * stand for at the table (red meat)?

Line 265- At the end of the table please write a legend for the abbreviations used on the table.

Page 9

I am concerned about the “lack of physical activity (PA)” – having 3 days of PA during wk?

Why the supplementary tables are not within the manuscript? I think they are quite important to appear in the manuscript because they respond to one of the objectives and should not be presented as supplementary material. 

Discussion

Page 9

Line 322 and 324 – “are” should be substituted for  “were”

Line 324 – Please add the SELF-REPORTED to the sentence:  “significantly associated with decreased consumption of vegetables, objective long sleep duration and SELF-REPORTED lack of physical activity”

Line 325 – Objective long sleep duration: what does it mean? You did not specify it anywhere in the manuscript. This is confusing.

>450 mins of TST is a long sleep duration? This is >7,5h, and according to National Sleep Foundation between 7- 8 is the amount recommended for this age range, being from 5 – 9h accepted. How do you explain it?

Line 368 – The reference cited [60] does not investigate sleep, is it correct? Ross R. The pathogenesis of atherosclerosis: a perspective for the 1990s. Nature. 1993 Apr 29;362(6423):801-9. doi: 10.1038/362801a0. PMID: 8479518.

Although it's not the objective of the manuscript the mechanisms of which long sleep duration (>9h) is related to increased inflammation should be discussed and referred.

Conclusion

Page 10

Line 382-386 is discrepant from the findings: exercise? Insulin resistance? 12 wk exercise program? Improvement of cognition? In the conclusion please do not cite references

Limitation

Page 10

Line 397 – using actigraphy instead of PSG is not a limitation. Please exclude. Different exams to answer different questions.

Thank you and hope these considerations add to the manuscript.

Author Response

please, see the attachment

Reviewer 2 Report

Thank you for the opportunity to review this manuscript titled, “Poor Diet, long sleep, and lack of physical activity are associated with inflammation among non-demented community-dwelling elderly.” Please see comments below:

  • Please provide a rationale for including participants with MCI and CNI in this analysis. From the title, it appears that you are including “non-demented” elderly participants, but in the methods you include MCI. It would be worth clarifying how MCI is different from “demented.” Also, from the methods, it seems like you are comparing predictors between MCI and CNI groups, however, the analysis combines the groups.
  • Some sentences in the introduction are missing citations, please correct.
  • Methods section is nicely written.
  • Please define CNI the 1st time it’s used in the manuscript.
  • For clarification, of the 505 total participants in phase II, some were diagnosed with MCI despite being in the MMSE>24 group? This wasn’t clear in the methods.
  • For the section on sleep measurement in the methods section, is it the case that you only considered long sleep vs everything else? As short sleep also contributes to inflammation, you should note the proportion of participants with short sleep, or analyze this group separately. Some recognition of short sleep duration is needed.
  • For Tables 1 and 2, add columns for not only the total sample, but also for the CNI and MCI groups. Then you wouldn’t need additional tables to display this information and could be viewed all together. You can combine mean and SD into 1 column.
  • In the discussion, you state the source population of your study consumes a large amount of vegetables. Can you comment on how the consumption of your sample compares to other cohorts, and how this plays into inflammation? For example, although inflammation was associated with vegetable consumption in your sample, was the overall inflammation lower compared to other studies, or overall vegetable consumption higher? Also, do you think the significant associations you found are clinically relevant? It looks like a very small difference in vegetable consumption according to the betas in the multivariate model.  
  • The limitations section focuses mainly on the actigraphy measurements. Please also add limitations for other considerations. For example, other factors, such as medications or medical diagnosis, which may affect inflammation, sleep, and diet, were not included. Also, when assessing multiple correlations, such as for diet, it’s likely significant correlations will be found.

Author Response

please, see the attachment

Reviewer 3 Report

Overview:

The paper summarise an experiment looking at the association of some life-style parameters with markers of inflammation in the elderly with and without MCI.

There are a number of issues that need to be addressed:

  1. Introduction:
    1. it would be good to include a hypothesis or, if not, justify why.
    2. Missing references on lines 33 and 38
    3. Line 55-57 – there seem to be many such studies with just a cursory search, perhaps the claim needs to be made more specific or reduced in scope. This seems to be the same problem with lines 68-69 and 76-77.
  2. Methods:
    1. Line 101 – it is unclear how the threshold was set, please clarify
    2. Figure 1 is potentially a bit misleading with the numbers in the MMSE > 24 group as there were more but they were excluded – also the figure has no title
    3. Line 211 – should have a reference
    4. Physical activity measure – it is unclear how this was derived and if it has been validated. Please expand and provide references.
    5. Due to the large number of potential multiple comparisons some form of correction should be applied or a justification of why one isn’t needed spelt out.
  3. Results:
    1. Where possible report effect sizes rather than just p values which don’t give much information about what is going on
    2. For Table 1 it would be good to see the values split by group as well
    3. Please explain how the TST>450min threshold was derived.
  4. Discussion:
    1. Report limitations of EDS, subjective sleep duration, these are not good measures as can be seen in table 2 where EDS is much lower than expected and TST is very different to the objective measure
    2. Line 379-382 – unclear sentence – please rephrase, also I don’t think this conclusion can be drawn from this study - justify
  5. Implications:
    1. Line 414-415 – it is not clear how the conclusion about long sleep being caused by inflammation is determined – please expand and supply references. Further it is not clear how the advice to avoid sleep prolonging medication logically follows from what comes before given the proposed direction of action – please explain or rephrase

Author Response

please, see the attachment

Round 2

Reviewer 1 Report

Thanks for the revised version of the manuscript. The authors have adequately addressed my prior concerns. The manuscript has improved and it is suitable for publication.